# Enhancing the Performance of Organic Phototransistors Based on Oriented Floating Films of P3HT Assisted by Al-Island Deposition

**DOI:** 10.3390/ma16155249

**Published:** 2023-07-26

**Authors:** Tejswini K. Lahane, Shubham Sharma, Moulika Desu, Yoshito Ando, Shyam S. Pandey, Vipul Singh

**Affiliations:** 1Molecular and Nanoelectronics Research Group (MNRG), Department of Electrical Engineering, IIT Indore, Indore 453552, Madhya Pradesh, India; phd1801202006@iiti.ac.in; 2Graduate School of Life Science and System Engineering, Kyushu Institute of Technology, 2-4, Hibikino, Wakamatsu, Kitakyushu 808-0196, Japan; sharma.shubham457@mail.kyutech.jp (S.S.); moulika.d2009@gmail.com (M.D.); yando@life.kyutech.ac.jp (Y.A.)

**Keywords:** organic phototransistor, P3HT, photosensitivity, Al-island, UFTM

## Abstract

The fabrication of high-performance Organic Phototransistors (OPTs) by depositing Al-islands atop Poly(3-hexylthiophene) (P3HT) thin film coated using the unidirectional floating-film transfer method (UFTM) has been realized. Further, the effect of Al-island thickness on the OPTs’ performance has been intensively investigated using X-ray photoelectron spectroscopy, X-ray Diffraction, Atomic force microscopy and UV-Vis spectroscopy analysis. Under the optimized conditions, OPTs’ mobility and on–off ratio were found to be 2 × 10^−2^ cm^2^ V^−1^ s^−1^ and 3 × 10^4^, respectively. Further, the device exhibited high photosensitivity of 10^5^, responsivity of 339 A/W, detectivity of 3 × 10^14^ Jones, and external quantum efficiency of 7.8 × 10^3^% when illuminated with a 525 nm LED laser (0.3 mW/cm^2^).

## 1. Introduction

Phototransistors have emerged as efficient devices for converting optical signals into electrical signals with amplified output, offering high sensitivity and low noise levels. These devices play a critical role in a wide range of applications, including imaging systems, communication technologies, and environmental and optical sensing [1,2,3,4,5]. In contrast to conventional phototransistors based on inorganic semiconductors, organic semiconductor-based phototransistors have garnered considerable attention due to their advantageous features, such as efficient light absorption, solution processability, and flexibility. Organic semiconducting materials have emerged as promising materials for wearable sensors, particularly in real-time health monitoring and imaging systems [6,7,8,9]. Their unique properties, coupled with the ability to fine-tune their electronic characteristics through rational molecular design, make them highly suitable for opto-sensing applications across a broad spectrum ranging from ultraviolet (UV) to near-infrared (NIR) [9,10,11]. Extensive research has been dedicated to exploring a wide range of organic materials for their potential use in visible light photodetection. P3HT is a well-researched polymer semiconductor with a high absorption coefficient in the visible light wavelength region. Single-organic-layer, bulk-heterojunction, and planar-heterojunction thin-film device structures are frequently utilized and studied for the fabrication of phototransistors due to their simpler and more cost-effective manufacturing conditions [12]. In single-organic-layer OPTs, the active semiconductors are primarily electron acceptors or donors. However, the single organic layers used in OPTs often have low photoresponsivity. This limitation arises due to two main factors: the relatively low charge carrier mobility exhibited by the active semiconductor layer and the inefficient dissociation of excitons within the layer [12]. By employing the drop-casting technique, Pal et al. fabricated P3HT-based OPTs with a photosensitivity of 3.8 × 10^3^ [13]. Zhang et al. reported OPTs fabricated using P3HT with a responsivity of 26.38 mA/W under 600 nm light illumination [14]. In order to enhance the optical properties, considerable efforts have been made to improve the morphology of photodetectors by incorporating polymer nanowires and quantum dots. This deliberate modification resulted in a significant enhancement of the devices’ sensitivity and responsivity. Wouter Dierckx et al. achieved a responsivity value of 250 A W^−1^ using P3HT nanofibers [15]. Additionally, a P3HT single-crystal nanowire OPT was also reported with a photosensitivity of 1 × 10^3^ [16]. The incorporation of these nanostructures, however, adds complexity to the fabrication process. Further, the hybrid bulk-heterojunction-based OPTs fabricated by Xu et al. were found to have very high photosensitivity of the order of 10^4^ [17]. Bulk-heterojunction structures, however, affect the overall film homogeneity and are difficult to reproduce. Nevertheless, it should be noted that the incorporation of such hybrid structures poses challenges in terms of achieving film homogeneity and reproducibility, thus limiting their practical application and scalability. On the other hand, a single-polymer thin film gives more reproducibility and can be easily processed. In recent years, significant efforts have been dedicated to improving polymer thin films’ performance by employing various methods to orient the polymer chains in a unidirectional fashion during the film fabrication process. By using the floating-film transfer method (FTM), Bhargava and Singh reported high photosensitivity of 10^4^ using oriented P3HT [18]. However, it is worth mentioning here that the reported performance of organic phototransistors (OPTs) still requires significant improvement for their practical applications. Despite their intrinsic benefits, OPTs have trouble obtaining the ideal optical performance characteristics, such as high responsivity, photosensitivity, detectivity, and EQE. In order to improve the optical performance of OPTs, the exciton dissociation and dark-current reduction are the key matrices. To effectively lower noise and improve the signal-to-noise ratio of OPTs, dark currents must be effectively reduced. Additionally, efficient exciton dissociation is imperative for maximizing the generation of photo-induced charge carriers, leading to improved photoresponsivity and overall device performance.

In this study, OPTs in the bottom-gate top-contact device architecture were fabricated using P3HT as an active layer. A unidirectional floating-film transfer method (UFTM), developed and improvised in our laboratory, was used to fabricate the uniform and oriented thin film of P3HT, and a very thin layer of Al-islands was deposited in the channel region. This approach was implemented to effectively reduce the dark current and enhance the exciton dissociation. Moreover, the influence of varying the Al-island thicknesses on the performance of the fabricated OPTs was comprehensively examined and, the Al-island thickness was optimized to 5 nm. The OPT with a 5 nm Al-island layer exhibited a substantial fourfold increase in photosensitivity compared to the pristine device. The optimized device demonstrated an impressive on–off ratio of 4 × 10^4^, with a photosensitivity of the order of 2 × 10^5^ and a responsivity of 339 A W^−1^.

## 2. Material and Methods

### 2.1. Materials 

Regio-regular P3HT (RR-P3HT) with an average molecular weight (M_w_) of 50,000–75,000, ethylene glycol (Eg), glycerol (Gl), Toluene, Trichloro(octadecyl)silane (OTS), chloroform (anhydrous) were purchased from Sigma Aldrich (Tokyo, Japan) and used as received. Acetone and iso-propyl alcohol (IPA) were purchased from the Wako Chemical Company, Tokyo, Japan. 

### 2.2. Device Fabrication and Thin-Film Preparation

Highly p-doped silicon substrates with a 300 nm thermally grown silicon dioxide (SiO_2_) layer and an aerial capacitance (C_ox_) of 10 nFcm^−2^ were utilized as the gate electrode and gate dielectric for the fabrication of OPTs. Substrates (1 × 1 cm) were cleaned in ultrasonic baths of acetone, isopropanol, and chloroform for 10 min each and annealed at 150 °C for 1 h. Thereafter, the samples were kept under UV-ozone treatment for 15 min to make the substrate hydrophilic for better surface adhesion to OTS treatment. Later, the substrates were placed in a 10 mM OTS solution in dehydrated toluene for 4 h to create a self-assembled monolayer of OTS to make the SiO_2_ surface hydrophobic. After that, a sample was cleaned ultrasonically in toluene and annealed at 100 °C for 30 min to remove the residual solvents on the surface. An amount of 30 mg/mL of P3HT solution prepared in super-dehydrated chloroform (Wako, Tokyo, Japan) was utilized to fabricate the oriented thin films using the UFTM. In this technique, 10 µL of solution was dropped onto the Eg:Gl (3:1) liquid substrate at 50 °C. The polymer solution flowed over the underlying liquid substrate, and at the same time, the polymer solvent evaporated, resulting in a thin solid film floating over the liquid substrate. Due to the viscous force applied by the underlying liquid substrate, the floating film was oriented perpendicular to the flow direction [19]. Further, a polarizing sheet was utilized to identify the orientated portion of the film, and the oriented thin film was cast onto the substrate. Afterward, we annealed the sample in a glove box at 100° C for 10 min. The silver source and drain electrodes were deposited using physical vapor deposition with a 20 μm channel length and 2 mm channel width of the device. Later, Al was deposited on the device channel with varying thicknesses of 2 nm, 3 nm, and 5 nm. Figure 1 shows the schematic device structure of the fabricated organic phototransistor using the SiO_2_ gate dielectric and Al-island deposition. For the optical measurements and surface analysis, the P3HT thin film was deposited on a cleaned glass substrate incorporating Al deposition with varying thicknesses, as discussed above.

### 2.3. Characterization

Polarized electronic absorption spectral measurements were performed using a JASCO V570 spectrophotometer (JASCO, Tokyo, Japan) equipped with a Glan–Thomson prism. Electrical characterization was conducted using a computer-controlled 2-channel source-measure unit (Keithley-2612, Tektronix, Tokyo, Japan). In order to measure the photo-response of the devices, they were illuminated with the monochromatic green light at a wavelength of 525 nm, with an incident power of 0.3 mW cm^−2^. The crystallinity of the P3HT thin film was assessed through out-of-plane X-ray diffraction (XRD) utilizing a Cu-K_α_ source with a wavelength of 1.5406 Å and a 2θ range of 5°–30°. For X-ray photoelectron spectroscopic (XPS) analysis, an XPS012A (Shimadzu, Kyoto, Japan) was used. Additionally, atomic force microscopy (AFM) images of the P3HT thin film with Al-islands were captured in tapping mode using an AFM5300E instrument by Hitachi High Tech Co., Tokyo, Japan. 

## 3. Results and Discussion

### 3.1. Thin-Film Characterization

AFM measurements were carried out to visualize the surface morphology of the P3HT films coated with different thicknesses of Al deposition using thermal evaporation and the results are shown in Figure 2a–d. The AFM image of the P3HT thin film using the UFTM (Figure 2a) shows well-oriented domains with sizes of up to a few micrometers. These oriented domains aligned parallel to the polymer orientation direction. Further, Figure 2b–d shows small Al-islands of around ~100–200 nm in width. The number of Al-islands on the surface increased with the increasing thickness of the Al deposition. For instance, in the case of the 5 nm Al-island thickness, well-grown particles with 150–200 nm diameters covered the entire surface without forming a continuous conducting path. The root mean square (RMS) roughness values observed for 2 nm, 3 nm, and 5 nm deposited Al-islands were 0.5 nm, 0.7 nm, and 1 nm, respectively. Due to the fact that metal deposits appeared in tiny molecular clusters during thermal evaporation, the minimization of the surface energy may have aided in the development of the islands during this process [20].

Further, the chemical interaction between Al and P3HT was investigated through XPS analysis, as illustrated in Figure 3. To gain further insights into the aluminum and carbon bonding, the Al 2p XPS spectrum was examined. Figure 3a shows the peaks observed in the Al 2p XPS spectra of the P3HT thin films with varying thicknesses of Al. In the spectra, a peak appeared at approximately 71 eV, indicating the presence of Al. For the P3HT films with a deposition of 3 nm Al, a distinct peak related to aluminum-oxygen-carbon (Al-O-C) bonds was clearly observed at 72.9 eV (Figure 3b). The appearance of an Al-O-C peak suggests the presence of aluminum oxide on the surface of P3HT. This aluminum oxide adsorbed and interacted with the carbon (C) atoms in the alkyl chain of edge-on-oriented P3HT, forming an Al-O-C bond. However, compared to the Al 2P peak, the intensity of the Al-O-C peak was significantly lower, which suggests the feeble presence of Al-O-C bonds. Hence, the formation of Al-O-C does not have an appreciable impact on the device performance [21,22]. When examining the oxygen (O) 1s peak (Figure 3c), it can be observed that the overall proportions increased after the deposition of Al. Further deconvolution of the O 1s peak showed a small peak at 531 eV. This peak corresponds to the Al-O peaks, as shown in Figure 3d. When Al is deposited on the P3HT surface, it can react with the residual oxygen in the evaporation chamber, leading to the formation of aluminum oxide [20]. The peak at 550 eV is the satellite peak of Al metal. This peak appears to be due to background photoelectron emission [23].

In Figure 3f, the intensity of the S 2p peak corresponding to sulfur reduced after the deposition of Al. The enhanced O penetration can result in an increased occupancy of space by P3HT, leading to a reduction in the sulfur (S) content. Al has a high electron affinity to oxygen and a much smaller affinity to sulfur [21,22]. Furthermore, for the 0 nm, 2 nm, and 3 nm thick Al deposited over P3HT, we observed systematic shifts in the S 2P peaks toward higher binding energy, which can be attributed to the downward band bending in P3HT due to the transfer of an electron from aluminum to P3HT [24,25]. Therefore, the increased Al thickness resulted in higher band bending and a wider depletion layer in P3HT. Also, there was an obvious increase in the Al 2p and O 1s peaks, and we believe that the 5 nm Al/P3HT films should show similar characteristics. Further, the overall area of the carbon (C) 1s peak decreased with Al deposition (Figure 3e). This observation suggests the possibility of the Al bonding with carbon molecules present in P3HT. The reduction in the C 1s peak area is indicative of the formation of Al-carbon bonds [21,22].

Next, the polarized electronic absorption spectral analysis and microstructural characterization by (XRD) are illustrated in Figure 4. The absorption spectra for P3HT with varying thicknesses of Al-islands are shown in Figure 4a. P3HT showed an absorption peak with λ_max_ at 514 nm, along with the vibronic shoulders appearing at 550 nm and 600 nm for 0-1 and 0-0 vibrational transitions [26]. It was observed that there was no significant shift in the absorption peak of P3HT after varying the thickness of the deposited Al-islands. However, increasing the thickness of the Al-islands would cause light scattering; hence, the absorption of P3HT was slightly lower for the 5 nm Al. Additionally, polarized absorption spectra were used to analyze the orientation characteristics of the UFTM-coated P3HT films (as shown in Figure 4b). Linearly polarized light was generated using a Glan–Thomson prism and directed toward the samples in a controlled manner. The plane of polarization of the incident light was systematically adjusted to be parallel and perpendicular to the orientation direction of the polymer chains in the films. This approach allowed for precise and quantitative measurement of the extent of the polymer orientation in the samples. The dichroic ratio (DR) was calculated using Equation (1).
(1)$$DR=\frac{Maximum\;absorption\;at\;\lambda max(\|)}{Absorption\;at\;\lambda max(⊥)}$$

The calculated optical DR for the UFTM-coated P3HT thin films was 1.9. The small value of the measured DR for RR-P3HT can be attributed to its high crystallinity, as observed by us previously [24].

Further, Figure 4c shows the out-of-plane XRD patterns of the P3HT thin film with varying Al-island thicknesses. For all the Al-island-deposited P3HT thin films, sharp diffraction (h00) peaks related to lamellar stacking of alkyl side chains were observed up to the third order at 2θ of 5.3°, 10.6°, and 15.9° [27]. The clear diffraction peaks show that the polymer thin films produced using the UFTM not only had a high degree of crystallinity but also appreciably good molecular ordering, as indicated by the polarized absorption spectra. The Scherrer–Debye equation was utilized to calculate the quantitative measurement of the grain sizes of the P3HT film [28].
(2)L=0.9λ/β cosθ
where L represents the crystallite grain size, β represents the full width at half maximum (FWHM) for the (h00) diffraction peak, and λ represents the wavelength of the incident X-ray beam used for the measurement. The calculated grain size of the P3HT film corresponding to the (100) diffraction peak was found to be 18 nm, which was not affected by the Al-islands.

### 3.2. Device Characterization

Lastly, the Al-islands deposited on the thin film and their electronic implications on the charge transport in the OPTs were studied. Figure 5 illustrates the current-voltage (I–V) characteristics for the fabricated devices using pristine and oriented RR-P3HT thin films without Al-island deposition. The devices clearly exhibit a typical p-type OFET characteristic. Figure 5a shows the output characteristics of the OPT (under dark conditions) fabricated using P3HT films without Al-island deposition. The modulation effect caused by the gate-to-source voltage (V_GS_) is clearly seen in the output characteristics, with strong linearity for low V_DS_ and saturation characteristics for high V_DS_. Furthermore, the transfer characteristics were evaluated by sweeping the V_GS_ across a range from 80 V to −80 V while maintaining a fixed V_DS_ at −80 V, as shown in Figure 5b. The on–off ratio of the device was 2 × 10^3^, with a threshold voltage (*V_th_*) of 20 V. Under dark conditions, the mobility (*μ_sat_*) of the OPT in the saturation regime was calculated using Equation (3) [12].
(3)$$I_{DS}=\frac{\mu_{sat}CiW}{2L}(v_{GS}-v_{th}{)}^{2}$$
where *µ_sat_* is the saturated field-effect mobility, I_ds_ is the drain-source current, V_GS_ is the gate voltage, and *V_th_* is the threshold voltage. C_i_ is the aerial capacitance of the device, which is 10 nFcm^−2^. The *μ_sat_* was calculated using the slope of the |I_DS_|^½^ vs. V_GS_ plot in the saturated regime. The *μ_sat_* for the UFTM-coated thin films without Al-island deposition in the dark was estimated to be 2 × 10^−2^ cm^2^ V^−1^ s^−1^.

Next, we deposited Al-islands with different thicknesses on the channel region of P3HT thin films. Upon the deposition of Al-islands, a Schottky junction was formed at the contact interface between Al and P3HT. This can be attributed to the transfer of electrons from Al to P3HT, leading to the depletion of holes in the P3HT layer. In the past, Singh et al. utilized the photoluminescence (PL) quenching method and determined that the depletion-layer width in Al/P3HT/ITO diodes was approximately 17 nm [29]. In the device configuration studied here, the deposition of a thin Al-island layer in the channel region resulted in a reduction in the channel width and conductance, which can be attributed to the depletion of the P3HT layer, as schematically illustrated in Figure 6b. The impact of varying the thicknesses of the Al-island deposition on the performance of OPTs is shown in Figure 6c,d. The output characteristics of the OPTs with different thicknesses of Al-island deposition are depicted in Figure 6c. The measurements were conducted under dark conditions at the cut-off region with V_GS_ = 0. It can be observed that as the thickness of the Al-island layer increased (0, 2, 3, and 5 nm), the dark current decreased. Increasing the Al-island thickness increased the depletion layer, thus decreasing the channel width, which can be attributed to suppressing the dark current. A minimum dark current was observed for the 5 nm Al-islands. Further increasing the thickness of the Al-island deposition to greater than 5 nm led to the formation of a uniform, conductive pathway in the channel region. Consequently, the device encountered a short-circuit condition if the Al-island thickness exceeded 5 nm. Further, to obtain the photo-response of the OPTs fabricated in the same way, the devices were illuminated with light of a wavelength of 525 nm and an intensity of 0.3 mW/cm^2^. Figure 6d shows the transfer characteristics of the corresponding OPT before and after light illumination. It can be seen that after Al-island deposition, the dark current and V_th_ decreased. A minimum dark current of 1.8 × 10^−9^ A was observed for the 5 nm island deposition. Further, after illumination, the transfer curves of all OPTs showed an increase in the I_DS_ and a positive shift in the V_th_ (shown in Figure 6d). The photovoltaic effect was dominant when the transistor operated in accumulation mode, i.e., V_GS_ < V_th_, which resulted in the shift of V_th_ toward a positive bias, as shown in Figure 6b. In this mode, when the semiconductor absorbed light, excitons were generated, which were later dissociated into free electron-hole charge carriers due to the applied negative bias. After that, the holes flowed toward the drain electrode, and the electrons accumulated under the source electrode. Due to the accumulation of electrons, the hole injection barrier between the semiconductor and the source decreased. As a result, the contact resistance decreased, V_th_ swung toward a positive value, and I_DS_ increased. Herein, we observed an increase in the photocurrent for the OPT incorporating Al-islands compared to the OPT without Al-islands. This enhancement can be attributed to the formation of a depletion layer at the interface between the P3HT and Al-islands. The presence of this depletion layer facilitated the facile dissociation of a greater number of excitons, primarily due to the built-in electric field at the interface, resulting in an enhancement of the photocurrent.

The variation in the on–off ratio and *μ_sat_* of the OPTs with different thicknesses of Al-islands is illustrated in Figure 7a. It can be observed that the OPT with a 5 nm Al layer exhibits the highest on–off ratio. This is due to the significant reduction in the dark current compared to the pristine devices. The *μ_sat_* of all OPTs remained at approximately 10^−2^ cm^2^ V^−1^ s^−1^. Furthermore, to quantitatively evaluate the optical performance of the fabricated OPTs, several key parameters were calculated, namely photosensitivity, photoresponsivity (R), specific detectivity (D), and external quantum efficiency (EQE). These parameters provide important insights into the OPTs’ ability to detect and convert incident light into an electrical signal.
(4)$$Photosensitivity=\frac{I_{ph}-I_{dark}}{I_{dark}}$$
where *I_ph_* is the drain photocurrent, and *I_dark_* is the drain dark current.

Figure 7b illustrates the photosensitivity plot of the OPTs as a function of the variable V_GS_. The maximum photosensitivity was observed in the off-state at a V_th_. However, the photosensitivity gradually decreased as the transistor entered the strong accumulation regime (V_GS_ = −80 V). This can be attributed to the increasing concentration of gate-induced carriers over the photo-generated carriers in the transistor channel. The obtained photosensitivity values for OPTs with Al-island thicknesses of 0, 2, 3, and 5 nm were 240, ~10^3^, ~2 × 10^3^, and ~2 × 10^5^, respectively. Notably, the OPT with a 5 nm Al-island thickness exhibited the highest photosensitivity, which was significantly greater than the previously published results for P3HT-based OPTs [18]. Another parameter, photoresponsivity (R), is a measure of the electrical response generated by the OPTs per unit of incident optical power. It was calculated using Equation (5).
(5)$$R=\frac{I_{ph}-I_{dark}}{P_{in}A}$$
where P is the incident light power intensity. Figure 7c shows the Photoresponsivity (R) vs. V_GS_ plot for all fabricated OPTs with different Al-island thicknesses. The value of R in all the OPTs increased from their off-state to on-state. The calculated R values for OPTs with Al-island thicknesses of 0, 2, 3, and 5 nm were 226, 497, 447, and 339 A W^−1^, respectively.

Specific detectivity (D) is a figure of merit that quantifies a device’s ability to detect weak optical signals while minimizing the impact of noise. It takes into account both the R and the noise characteristics of the device. A higher D indicates better sensitivity to low-level optical signals. In this case, the dominant contributor to the total noise was assumed to be the shot noise originating from the dark current [9,30,31]. D was calculated using Equation (6).
(6)$$D=R\sqrt{\frac{A}{2eI_{d}}}$$
where A is the device area (4 × 10^−4^ cm^2^ in this case). In our study, the D of the OPTs was enhanced 10 times by incorporating a 5 nm Al layer compared to the pristine OPTs. The external quantum efficiency (EQE) is a measure of a device’s ability to convert incident photons into electrical carriers. It represents the ratio of the number of charge carriers generated to the number of incident photons. The EQE provides insights into the efficiency of the light-to-electrical conversion process in the device and is given as EQE = hcR(eλ)^−1^, where h is the Plank’s constant, λ is the wavelength, c is the light velocity, and e is the electron charge [32]. The EQE for the OPT without Al-islands was 5.3 × 10^3^%, whereas, for the 5 nm thick Al-islands, it was 7.8 × 10^3^%. Table 1 presents a comprehensive summary of the device parameters. The OPTs integrated with Al layers demonstrated notable enhancements in terms of photosensitivity, R, D, and EQE. The thickness of the Al-islands played a pivotal role in enhancing the off-state characteristics and overall optical performance of the OPTs. In this device configuration, 5 nm was the optimal thickness of the Al-islands that achieved the highest photosensitivity and D.

Furthermore, Table 2 provides a brief comparison between this work and previously reported P3HT-based organic phototransistors (OPTs). Dierckx et al. achieved a three-order increase in photosensitivity by forming nanofibers of P3HT [15]. Aynehband et al. enhanced the photosensitivity of OPTs by incorporating perovskite nanocrystals blended with P3HT S [33]. Kim et al. attained a four-order sensitivity improvement by utilizing channel/dielectric/sensing triple layers [34]. Several techniques, such as blending, doping, and aggregation, have been previously explored to enhance the sensitivity of P3HT-based OPTs. Nevertheless, there remains a need for further improvements in the photosensitivity and responsivity of OPTs.

In comparison to the aforementioned studies, the P3HT-based OPT presented in this research demonstrates significantly higher photosensitivity of five orders and responsivity of 339 A/W. The obtained results exhibit immense promise and signify considerable potential for the advancement of OPTs that possess both high sensitivity and cost-effectiveness.

## 4. Conclusions

In summary, we successfully fabricated high-performance OPTs utilizing oriented thin films of P3HT in conjunction with Al-island deposition over the channel region. The incorporation of Al-islands in the device architecture effectively reduced the channel width by depleting the P3HT layer. The depletion of the P3HT layer resulted in improved device performance by minimizing unwanted current flow in the dark state and promoting efficient separation of excitons, leading to enhanced device sensitivity. Moreover, the influence of varying the Al-island thicknesses on the performance of the device was comprehensively examined, and the AI island was optimized to a thickness of 5 nm. These findings hold considerable promise for the advancement of high-performance photosensitive OPTs based on conducting polymers. The OPTs with a 5 nm Al layer exhibited a saturated field-effect mobility of 2 × 10^−2^ cm² V^−1^s^−1^ and an impressive on–off ratio of 3 × 10⁴ in dark conditions. Notably, these devices also exhibited outstanding photodetection performance with photosensitivity of 2 × 10^5^, photoresponsivity of 339 A W^−1^, photo detectivity of 3 × 10^14^ Jones, and EQE of 7.8 × 10^3^% under the irradiation of 0.3 mW cm^−2^ and 525 nm light. Thus, it can be concluded that the depleting channel width of an OPT through the incorporation of an optimized metal island is instrumental in optimizing the device’s performance and functionality.

## Figures and Tables

**Figure 1 materials-16-05249-f001:**
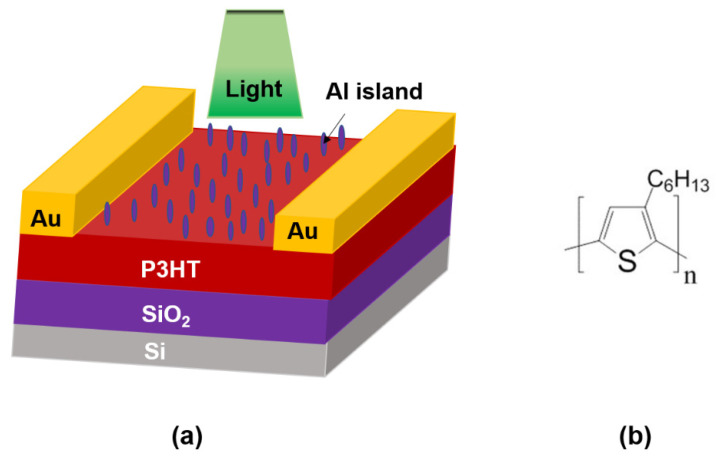
(**a**) Device schematic of the fabricated organic phototransistor using SiO_2_ gate dielectric and Al-island deposition, (**b**) P3HT structure.

**Figure 2 materials-16-05249-f002:**
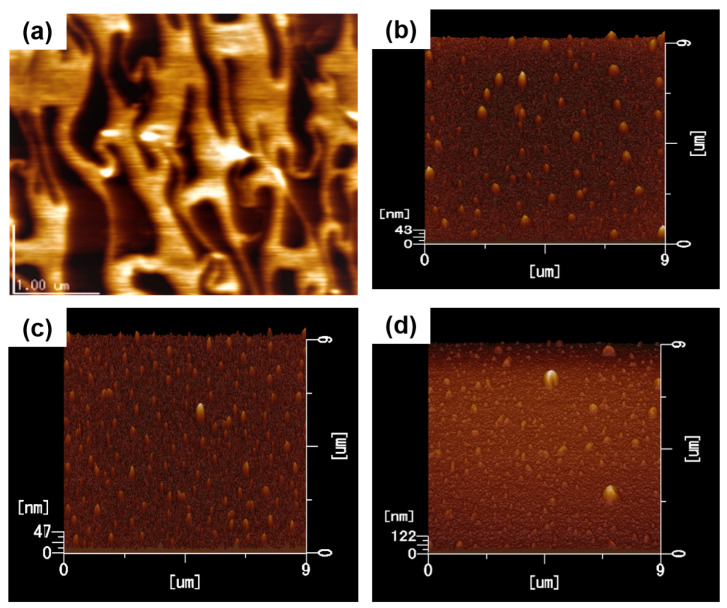
AFM images of P3HT thin film with (**a**) 0 nm, (**b**) 2 nm, (**c**) 3 nm, and (**d**) 5 nm Al-island deposition.

**Figure 3 materials-16-05249-f003:**
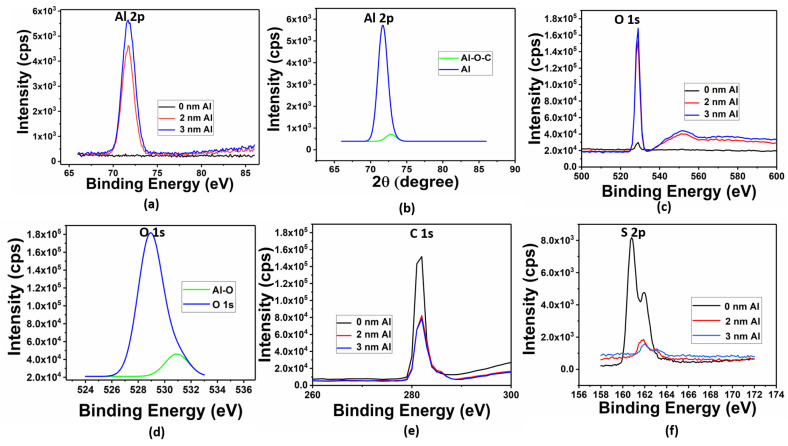
(**a**) Al 2p XPS spectra data, (**b**) Al 2p XPS fitting for 3 nm Al sample, (**c**) O1s XPS spectra, (**d**) O 1s fitting for 3 nm Al sample, (**e**) C 1s XPS spectra, (**f**) S 2p XPS spectra of P3HT thin film with different thickness of Al-island deposition.

**Figure 4 materials-16-05249-f004:**
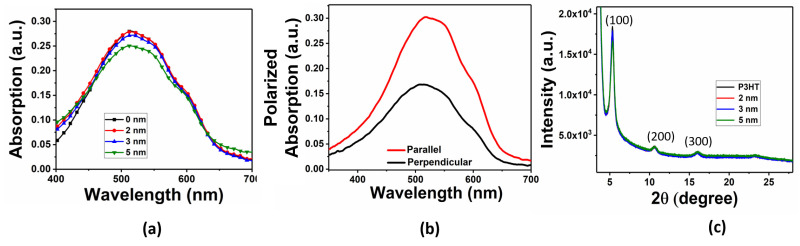
(**a**) Absorption spectra of UFTM-coated P3HT thin film with varying Al-island thicknesses, (**b**) Polarized absorption spectra of P3HT thin film, and (**c**) Out-of-plane XRD diffraction pattern of P3HT thin films with varying thicknesses of Al-island deposition.

**Figure 5 materials-16-05249-f005:**
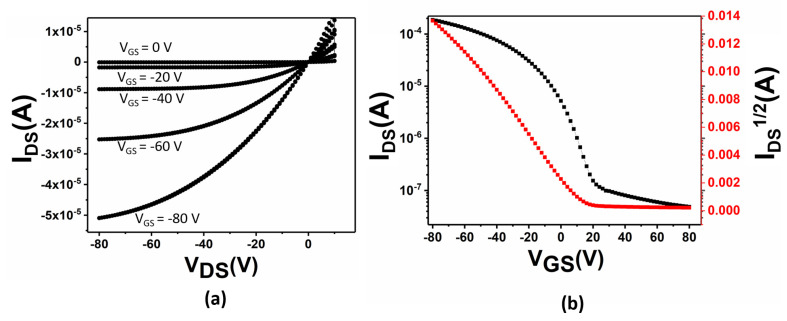
(**a**) Output characteristics, (**b**) Transfer characteristics and I_D_^0.5^ vs. V_GS_ plot of OPT using UFTM-coated P3HT thin film without Al-island deposition.

**Figure 6 materials-16-05249-f006:**
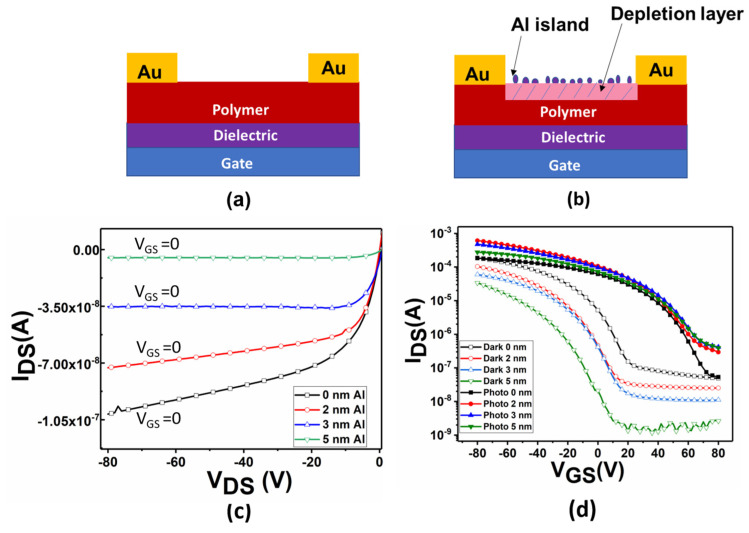
(**a**) Schematic of the device before Al-island deposition, (**b**) Depletion-layer formation after Al-island deposition, (**c**) Dark output characteristics at V_GS_ = 0, and (**d**) Transfer characteristics under dark and light illumination conditions.

**Figure 7 materials-16-05249-f007:**
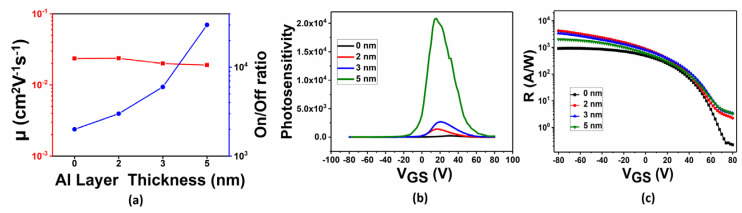
(**a**) Mobility and On/Off ratio vs. Al thickness variation, (**b**) Photosensitivity and (**c**) Photoresponsivity of P3HT with different thicknesses of the Al layer in terms of the changing V_GS_.

**Table 1 materials-16-05249-t001:** A summary of device parameters for OPTs with different thicknesses of Al deposition.

Parameter	0 nm Al	2 nm Al	3 nm Al	5 nm Al
μ_sat_ (cm^2^ V^−1^ s^−1^)	2.35 × 10^−2^	2.37 × 10^−2^	2.2 × 10^−2^	1.9 × 10^−2^
on–off ratio	2 × 10^3^	3 × 10^3^	6 × 10^3^	3 × 10^4^
off current (nA)	81	29	14	1.8
V_th_ (V)	21	16.2	16.6	13.5
photosensitivity	240	1 × 10^3^	2 × 10^3^	2 × 10^5^
R (A W^−1^)	226	497	447	339
D (Jones)	2 × 10^13^	8.4 × 10^13^	1 × 10^14^	3 × 10^14^
EQE (%)	53,271	117,150	105,364	78,257

**Table 2 materials-16-05249-t002:** Comparison of this work with previously reported OPTs.

Organic Semiconductor	Wavelength (nm), Light Intensity (mW cm^−2^)	μ_sat_ (cm^2^ V^−1^ s^−1^)	Photosensitivity	R (A W^−1^)	Ref.
P3HT	532, 0.4	7.22 × 10^−4^	10^4^	10	[18]
P3HT	630, 0.4	1.5 × 10^−3^	6.8 × 10^3^	250	[15]
P3HT blend	600, 100	7 × 10^−2^		21	[33]
P3HT	905, 3.8	3.2 × 10^−4^	10^4^	-	[34]
P3HT blend	300–700, 100	-	100	3	[35]
P3HT	525, 0.3	2 × 10^−2^	10^5^	339	This Work

## Data Availability

The data presented in this study are available on request from the corresponding author.

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
