# Peer review of "Enhancing the Performance of Organic Phototransistors Based on Oriented Floating Films of P3HT Assisted by Al-Island Deposition"

_materials, 2023, doi:10.3390/ma16155249_

Round 1

Reviewer 1 Report

1.       In the process of device preparation, what is the reason for annealing and UV-ozone treatment after cleaning the substrate?

2.       The full name and abbreviation of the time unit are mixed in the text. The "℃" in line 104 was incorrectly typed as "°".

3.       Lack of AFM measurement of the original surface compared to the deposited Al-island.

4.       The device with 5nm Al deposited in this paper has the best performance, but the XPS measurement for it is missing in Figure 3.

5.       The correlation peak of Al-O-C bond is observed in Figure 3(b). What are the causes of Al-O-C bond and its impact on the device?

6.       What is the cause of the peak at 550nm in Figure 3(c)?

7.       Change "Al2O3" to "Al-O" in Figure 3(d). Lack of discussion of Figure 3(d).

8.       In Figure 3(f), the reason for peak shift with the increase of Al thickness.

9.       The format of the figure is not uniform, and the horizontal coordinate font in Figure 3(d) is small. The horizontal coordinate "700" in Figure 4(a)(b) is not fully displayed. The longitudinal headings in Figure 6(d) are not completely displayed.

10.   In Figure 4(f), why does the XRD image of the film deposited with 2nm Al have a peak near 30nm compared with others?

11.   In Figure 5(a), the VGS  is not indicated. The figure of IDS1/2-VGS does not correspond to IDS-VGS in Figure 5(b).

12.   Delete "current" from "IDS current" in line 277.

13.   Delete "%" in row 9, column 2 of Table 1.

Author Response

Response to Reviewers' comments:

The authors are grateful to the reviewer for critically reviewing the manuscripts and providing us with comments/suggestions in the draft manuscript leading to its further improvisation. Please find below the point-wise rebuttal to your comment and suggestions.

Reviewer 1:

Comments to Author

Comment 1. In the process of device preparation, what is the reason for annealing and UV-ozone treatment after cleaning the substrate?

Response: The authors are thankful to the reviewer for the insightful comment. In the process of device preparation, annealing is used to remove residual solvents and UV-ozone treatment is used to make substrate hydrophilic for better surface adhesion to OTS treatment.

Corresponding change in manuscript: yes

Location: line no. 98, 102

Comment 2. The full name and abbreviation of the time unit are mixed in the text. The "℃" in line 104 was incorrectly typed as "°".

Response: We apologise for the mistake, the changes have been incorporated in revise manuscripts

Corresponding change in manuscript: yes

Location: line no. 104,97,100

Comment 3. Lack of AFM measurement of the original surface compared to the deposited Al-island.

Response: Authors are thankful to the reviewer for his valuable suggestion, changes have been incorporated in the revised manuscript.

Figure 2. AFM images of (a)Pristine P3HT thin film and P3HT thin film with (b) 2 nm (b) 3 nm, and(c) 5 nm Al-island deposition.

The respective text has also been added to the revised manuscript. The AFM image of P3HT thin film via UFTM (Fig 2 (a)) shows well-oriented domains with sizes up to a few micrometers. These oriented domains align parallel to the polymer orientation direction.

Corresponding change in manuscript: yes

Location: Figure 2(a), line no. 139-141

Comment 4. The device with 5nm Al deposited in this paper has the best performance, but the XPS measurement for it is missing in Figure 3.

Response: We are thankful to the reviewer for the comment. In the sequence of the experimentation, we fabricated the 0, 2, and 3nm Al/P3HT devices, where the 3nm Al/P3HT devices exhibited the best device performance. Later we sent these samples for the XPS measurement. Furthermore, we fabricated the devices using 5 and 10nm Al/P3HT as well to confirm the effect of Al barrier on the charge transport in P3HT. We found out that 5 nm showed an improved performance whereas 10 nm Al was short-circuited. There is an obvious increase in the intensity of Al 2P and O 1S peaks as the thickness of Al increases. However, in the XPS data of 2 nm and 3 nm thick Al/P3HT, we have observed that the XPS showed a similar characteristics, and measurement of 5nm Al/P3HT was not going to give any further information. Thus, we decided against the XPS measurement and analysis of 5nm Al/P3HT films. If the reviewer is of the opinion that the XPS of 5nm Al/P3HT is utmost necessary, we can provide the additional data. Further, we have revised the manuscript to make the XPS analysis clearer and the reason for not including the 5nm XPS data.

Corresponding change in manuscript: yes

Location : 178-182

Comment 5. The correlation peak of Al-O-C bond is observed in Figure 3(b). What are the causes of Al-O-C bond and its impact on the device?

Response :  The appearance of an Al-O-C peak suggests the presence of aluminium oxide on the surface of P3HT. This aluminium oxide adsorb and interact with the carbon (C) atoms in Alkyl side chain of edge-on oriented P3HT and form Al-O-C bond. However compare to Al 2P peak the intensity of Al-O-C peak was significantly low, which suggests feeble presence of  Al-O-C bonds. Hence formation of Al-O-C does not have appreciable impact on the device performance.

Corresponding change in manuscript: yes

Location: line no 160 to 165

Comment 6. What is the cause of the peak at 550 nm in Figure 3(c)?

Response : The peak at 550 eV is satellite peaks of Al. These peaks appear to the high binding energy sides to the main peak in the XPS spectrum. The appearance of satellite peaks in XPS can be attributed to various physical processes and interactions occurring during the photoemission process such as Shake-up and Shake-off Excitations, Augur electron decay and Plasmon Excitations. When a core-level electron is excited and ejected by an incident X-ray photon, energy is transferred to the remaining electrons in the atom. This energy transfer can lead to excitations of other electrons, causing them to be promoted to higher energy levels. The subsequent relaxation of these excited electrons can result in the emission of additional photoelectrons, leading to the observation of satellite peaks [1]

Ref:

http://www.qro.cinvestav.mx/~aherrera/reportesInternos/satellitesTransitionMetals.pdf

[1] N. Pauly, F. Yubero, and S. Tougaard, “Quantitative analysis of satellite structures in XPS spectra of gold and silver,” Applied Surface Science, vol. 383, pp. 317-323, 2016

Corresponding change in manuscript: yes

Location: line no 170 to 172

Comment 7. Change "Al2O3" to "Al-O" in Figure 3(d). Lack of discussion of Figure 3(d).

Response: Deconvolution of O 1s peak shows the small peak at 531 eV. This peak corresponds to Al-O peaks (as shown in Figure 3(d). When Al is deposited on the P3HT surface, it can react with the residual oxygen in the evaporation chamber, leading to the formation of aluminum oxide. Further, we have made the respective changes in the revised manuscript.

Corresponding change in manuscript: yes

Location: line no 167 to 171

Comment 8.  In Figure 3(f), the reason for peak shift with the increase of Al thickness.

Response : Shifting of S 2p peaks toward higher binding energies with increasing Al thickness are typically due to the downward band bending effect which is associated with electron transfer from aluminium into the P3HT [2, 3]. Therefore, increased Al thickness has resulted in higher band bending and a wider depletion layer in P3HT.

Ref:

[2]       X. Feng, L. Zhang, Y. Ye, Y. Han, Q. Xu, K.-J. Kim, K. Ihm, B. Kim, H. Bechtel, M. Martin, J. Guo, and J. Zhu, “Engineering the metal–organic interface by transferring a high-quality single layer graphene on top of organic materials,” Carbon, vol. 87, pp. 78-86, 2015.

[3]       X. Feng, W. Zhao, H. Ju, L. Zhang, Y. Ye, W. Zhang, and J. Zhu, “Electronic structures and chemical reactions at the interface between Li and regioregular poly (3-hexylthiophene),” Organic Electronics, vol. 13, no. 6, pp. 1060-1067, 2012.

Corresponding change in manuscript: yes

Location: line no 176 to 180

Comment 9.       The format of the figure is not uniform, and the horizontal coordinate font in Figure 3(d) is small. The horizontal coordinate "700" in Figure 4(a)(b) is not fully displayed. The longitudinal headings in Figure 6(d) are not completely displayed.

Response: As per the reviewer's suggestion, the changes has been incorporated in the revised manuscript.

Figure 3. (a) Al 2p XPS spectra data (b) Al 2p XPS fitting for 3 nm Al sample (c) O1s XPS spectra (d) O 1s fitting for 3 nm Al sample (e) C 1s XPS spectra (f) S 2p XPS spectra of P3HT thin film with different thickness of Al-island deposition.

Figure 4. (a) Absorption spectra of UFTM-coated P3HT thin film with varying Al-island thickness, (b) polarized absorption spectra of P3HT thin film and (c) out-of-plane XRD diffraction pattern of P3HT thin films with varying thickness of Al-island deposition.

Figure 6 (a) Schematic of device before, and (b) depletion layer formation after Al-island deposition (c) Dark output characteristics at VGS = 0, and (d) transfer characteristics under dark and light illumination conditions.

Corresponding change in manuscript: yes

Location: Figure 3(d), Figure 4(a)(b), Figure 6 (d)

Comment 10.   In Figure 4(f), why does the XRD image of the film deposited with 2nm Al have a peak near 30nm compared with others?

Response : We are thankful to the reviewer for the comment. The observed peak near 30 nm is instrumental error, which we have observed in the past as well. Hence, to avoid readers confusion we are showing plot only upto desired peak.

Corresponding change in manuscript: yes

Location: Figure 4 (f)

Comment 11.   In Figure 5(a), the VGS is not indicated. The figure of IDS1/2-VGS does not correspond to IDS-VGS in Figure 5(b).

Response : Authors are thankful to the reviewer for his valuable suggestion, changes has been incorporated in the revised manuscript in Figure 5(a-b).

Figure 5. (a) Output (b) Transfer characteristics and |ID0.5-VGS| plot of OPT using UFTM-coated P3HT thin film without Al-island deposition.

Corresponding change in manuscript: yes

Location: Figure 5 (a-b)

Comment 12.   Delete "current" from "IDS current" in line 277.

Response : As per the reviewer's suggestion, the changes has been incorporated in the revised manuscript.

Corresponding change in manuscript: yes

Location: line 281

Comment 13.   Delete "%" in row 9, column 2 of Table 1.

Response : As per the reviewer's suggestion, the changes has been incorporated in the revised manuscript.

Corresponding change in manuscript: yes

Location: Table 1

Reviewer 2 Report

see the attachment

Author Response

Response to Reviewers' comments:

The authors are grateful to the reviewer for critically reviewing the manuscripts and providing us with comments/suggestions in the draft manuscript leading to its further improvisation. Please find below the point-wise rebuttal to your comment and suggestions.

Reviewer 2 comments:

Comment 1: The abstract is recommended to be rewritten to more clearly reflect some aspects related to the study results.

Response: We are thankful the reviewer for the insightful comment. The abstract has been modified to make the study and results clear.

Abstract:  The fabrication of high-performance OPT by depositing Al islands atop the Poly(3-hexylthiophene) (P3HT) thin film coated by the unidirectional floating film transfer method has been realized. Further, the effect of Al-island thickness on the OPT performance has been intensively investigated using XPS, AFM, XRD and UV-Vis analysis. Under the optimized conditions, OPTs’ mobility and on-off ratio were found to be 2 × 10-2 cm2V-1s-1 and 3 × 104, respectively. Further, the device exhibited high photosensitivity of 105, responsivity of 339A/W, detectivity of 3 × 1014 Jones and EQE of 7.8× 103 %, when illuminated with 525 nm light LED laser (0.3 mW/cm2).

Corresponding change in manuscript: yes

Location: Abstract

Comment 2: For Organic Phototransistors, the abbreviations (OPTs) are used twice in the text, while for Poly(3-hexylthiophene) only the abbreviation is used. The measurement units must also be correlated „found to be 2 × 10-2 cm2V-1s-1 and 3 × 104. Further, the device exhibited”; - It is recommended to correct some typing errors such as: 100°; x-ray photoelectron…; - Table header 1 should be rewritten to more clearly reflect the data contained (2nm, 3nm?.)

Response: Authors are thankful to the reviewer for his valuable suggestion, changes have been incorporated in the revised manuscript

Corresponding change in manuscript: yes

Location:  Abstract, line no 11to17 101,130 and Table 1

Table 1:

Table 1. A summary of device parameters for OPTs having different thickness Al deposition

Parameters

0 nm-Al

2 nm-Al

3 nm-Al

5 nm-Al

μsat (cm2V-1s-1)

2.35 ×10-2

2.37 ×10-2

2.2 ×10-2

1.9×10-2

on-off ratio

2×103

3×103

6×103

3×104

off current (nA)

81

29

14

1.8

Vth (V)

21

16.2

16.6

13.5

Photosensitivity

240

1×103

2×103

2×105

R (AW-1)

226

497

447

339

D (Jones)

2×1013

8.4×1013

1×1014

3×1014

EQE (%)

53271

117150

105364

78257

Comment 3. For Table 3, a more extensive description is required in the text of the work, even if there are reference data;.

Response: We have modified the revised manuscript with extensive description of the text of the works given in Table 2.

 Furthermore, Table 2 provides a brief comparison between the present study and previously reported P3HT-based Organic Phototransistors (OPTs). Dierckx et al. have achieved a three-order increase in photosensitivity by forming nanofibers of P3HT [15]. Aynehband et al. have enhanced the photosensitivity of OPTs by incorporating perovskite nanocrystals blended with P3HT S [33]. Kim et al. have attained a four-order sensitivity improvement by utilizing Channel/Dielectric/Sensing Triple Layers [34]. Several techniques, such as blending, doping, and aggregation, have been previously explored to enhance the sensitivity of P3HT-based OPTs. Nevertheless, there remains a need for further improvements in the photosensitivity and responsivity of OPTs. In comparison to the aforementioned studies, the OPT presented in this research demonstrates significantly higher photosensitivity of five orders and a responsivity of 339 A/W. The obtained results exhibit immense promise and signify considerable potential for the advancement of OPTs that possess both high sensitivity and cost-effectiveness.

Organic Semiconductor

Wavelength (nm), Light intensity (mWcm-2)

μsat (cm2V-1s-1)

Photosensitivity

R (AW-1)

Ref

P3HT

532, 

0.4

7.22 × 10− 4

104

10

[18]

P3HT

630, 0.4

1.5 × 10-3

6.8 × 103

250  

[15]

P3HT blend

    600, 100

7 × 10− 2

21

[33]

P3HT

905,3,8  

3.2 × 10− 4

104

-

[34]

P3HT blend

 300-700, 

100

-

100

3

[35]

P3HT

525,

0.3

1.9 × 10− 2

105

339

This Work

Corresponding change in manuscript: yes

Location:  line no 363 to 372, Table 2

Comment 4. Optionally, it is recommended to give the 3D AFM images that can lead to a wider characterization of the morphological properties

 Response: Authors are thankful to the reviewer for his valuable suggestion. AFM images for Al deposited P3HT are already in 3D form but we have use front angle as we have observed quantity of Al island are better visible in this angle.

Corresponding change in manuscript: No

Round 2

Reviewer 1 Report

Accept as is